# Indications and Limits of Surgery for Spinal Metastases Derived from Lung Cancer: A Single-Center Experience

**DOI:** 10.3390/diagnostics13122093

**Published:** 2023-06-16

**Authors:** Silvia Terzi, Federica Trentin, Cristiana Griffoni, Elisa Carretta, Stefano Bandiera, Cristina Ferrari, Fabio Vita, Alberto Righi, Margherita Maioli, Dario De Biase, Annalisa Monetta, Giovanni Barbanti Brodano, Gisberto Evangelisti, Marco Girolami, Valerio Pipola, Marco Gambarotti, Alessandro Gasbarrini

**Affiliations:** 1Department of Spine Surgery, IRCCS Istituto Ortopedico Rizzoli, 40136 Bologna, Italy; silvia.terzi@ior.it (S.T.); federicatrentin94@gmail.com (F.T.); stefano.bandiera@ior.it (S.B.); fabio.vita@ior.it (F.V.); annalisa.monetta@ior.it (A.M.); giovanni@barbantibrodano.com (G.B.B.); gisberto.evangelisti@ior.it (G.E.); marco.girolami@ior.it (M.G.); valerio.pipola@ior.it (V.P.); alessandro.gasbarrini@ior.it (A.G.); 2Department of Programming and Monitoring, IRCCS Istituto Ortopedico Rizzoli, 40136 Bologna, Italy; elisa.carretta@ior.it; 3Laboratory of Experimental Oncology, IRCCS Istituto Ortopedico Rizzoli, 40136 Bologna, Italy; cristina.ferrari@ior.it; 4Anatomy and Pathological Histology Unit, IRCCS Istituto Ortopedico Rizzoli, 40136 Bologna, Italy; alberto.righi@ior.it (A.R.); margherita.maioli@ior.it (M.M.); marco.gambarotti@ior.it (M.G.); 5Molecular Diagnostics Unit, Department of Pharmacy and Biotechnology, University of Bologna, 40136 Bologna, Italy; dario.debiase@unibo.it

**Keywords:** spinal metastases, non-small-cell lung cancer, spine surgery, survival, quality of life

## Abstract

Lung cancer is the second most frequently diagnosed cancer in the world, and surgery is an integral part of the treatment for spinal metastases. The aims of this retrospective study were to assess the overall survival of surgically treated patients affected by lung cancer spinal metastases and identify any factors related to a better survival rate. We recruited 56 consecutive patients (34 male and 22 female) surgically treated for metastatic lung cancer in the spine from 2009 to 2019. Surgical indications were based on a previously published and validated flow chart following a multidisciplinary evaluation. We assessed the localization of vertebral metastases, the presence of other bone or visceral metastases, neurological status according to the Frankel score, ambulatory autonomy, and general status, measured with the Karnofsky performance scale. The expected prognosis was retrospectively assessed according to the revised Tokuhashi score. The median survival was 8.1 months, with over a third of patients surviving more than 1 year. We observed a global improvement in all clinical parameters after surgical treatment. The Tokuhashi predictive score did not correlate with survival after surgery. The results of this study suggest that the surgical treatment of symptomatic spinal metastases from lung cancer can improve quality of life, even in patients with a shorter life expectancy, by controlling pain and improving autonomy.

## 1. Introduction

According to the International Agency for Research on Cancer, in 2020, lung cancer was surpassed by breast cancer as the most frequently diagnosed cancer in the world. However, with 1.8 million deaths, lung cancer remains the most frequent cause of death from malignant cancer, with extremely high human, social, and economic costs [1]. Lung cancer is an aggressive, typically systemic disease with a high propensity to metastasize at a distance. Among lung cancers, small-cell lung cancer (SCLC), also called microcytoma, is a separate group with its own characteristics. It has a particularly rapid course and is often disseminated at onset, with a poor prognosis in the short term and few opportunities for surgical treatment. Non-small-cell lung cancer (NSCLC), with its different histotypes, has a high affinity for bone and skeletal localization and is the most frequent cancer for NSCLC metastases. Among the skeletal localizations, the most frequent is vertebral [2,3]. Symptomatic vertebral metastases are disabling and significantly worsen the clinical picture. Skeletal-related events (SREs), such as pathological fractures, instability, intractable pain, and myeloradiculopathy, have negative impacts on quality of life and prognosis. In general, the median survival after a diagnosis of NSCLC is 13 months for non-metastatic disease and 5 months for metastatic disease [4]. Recent therapeutic advances have partially improved this prognosis. Immunotherapeutic drugs, including immune checkpoint inhibitors, have been effective in treating advanced NSCLC [5]. Additionally, some patients with epidermal growth factor receptor (EGFR) mutation respond well to targeted therapy with tyrosine kinase inhibitors [6]. Spinal metastasis surgery is an integral part of the treatment, which should always be addressed with a multidisciplinary approach. The purposes of this study were to assess the overall survival of surgically treated patients with NSCLC spinal metastases in order to evaluate the impact of surgery on the clinical picture and identify any factors related to a better survival rate.

## 2. Materials and Methods

This was a retrospective cohort study of 56 consecutive patients surgically treated for spinal metastases due to lung cancer from 2009 to 2019. The study was approved by the CE-AVEC Emilia Romagna Ethics Committee in June 2018 (protocol number CE-AVEC 284/2018/Oss/IOR). The signing of study-specific informed consent was not required for this retrospective study due to the regulations for health institutions dedicated to scientific research. Demographic, anamnestic, and clinical data were taken from electronic health records. For each patient, at baseline, we assessed the localization of vertebral metastases, the presence of other bone or visceral metastases, neurological status according to the Frankel score [7], ambulatory autonomy, and general status, measured with the Karnofsky performance scale [8]. The expected prognosis was retrospectively assessed according to the revised Tokuhashi score [9]. The surgical indication and type of surgical procedure were based on the previously published and validated algorithm for the management of spinal metastases [10,11]. Each therapeutic decision was made following the multidisciplinary evaluation of a team composed of an oncologist, a radiotherapist, and a spine surgeon (Figure 1).

We first assessed whether the patient was operable using the ASA Physical Status Classification (ASA score). Patients with a score equal to or greater than 4 were not eligible for surgery. Therefore, they received systemic and radiotherapy treatments and eventually vertebroplasty to reduce their pain in cases of vertebral collapse. Operable patients with an ASA score of 1 to 3 and symptoms of spinal cord compression underwent radiotherapy, chemotherapy, or hormone therapy according to the sensitivity of the tumor based on its histological type and, subsequently, surgical treatment, which could include decompression and stabilization for pathological fractures and debulking or en bloc resection in cases of isolated metastases, possibly associated with vertebroplasty to reduce pain. If the tumor’s histological type was not sensitive to non-surgical therapies, decompression and stabilization were performed.

All patients recruited for this study underwent spinal surgery. Posterior decompression was performed in most patients, either via laminectomy and stabilization or via debulking (intended as macroscopically large removal of metastasis with an intralesional margin, associated with stabilization). En bloc resection was performed in only one case, while four patients underwent minimally invasive stabilization procedures.

We collected information regarding non-surgical therapies (radiotherapy and systemic therapies) performed before and after surgery, intra- and postoperative complications, any re-operations, and local recurrence of the disease. The same clinical data collected at baseline were evaluated at discharge and at follow-up through outpatient visits or phone calls. In all cases where it was not possible to directly contact the patient, we operated through local health inspectors to find out whether and when the patient died. Additionally, molecular typing of the spinal metastases biopsies was performed via next-generation sequencing (NGS) using a laboratory-developed multi-gene panel that can detect EGFR mutations or other mutations [12].

### Statistical Analysis

The demographic and clinical characteristics of the study cohort were summarized using the median and range or absolute and percentage frequencies. The outcome of interest was overall survival (OS), defined as the time elapsed from the date of surgery to the date of death or to the date of the last follow-up. The Kaplan–Meier method was used to estimate OS, and the logrank test was used to evaluate differences in survival between potentially prognostic factors. All analyses were performed using SAS 9.4 (SAS Institute, Cary, NC, USA) software.

## 3. Results

Between November 2009 and April 2019, 56 patients (34 male and 22 female) affected by spinal metastases from lung cancer were surgically treated at our center. The patients’ ages ranged from 42 to 84 years, with a median of 65 years. All primary tumors were non-small-cell lung cancers (NSCLC). In most patients (67.9%), the lesion was localized in the thoracic spine. Over 70% of patients (37 patients) had a revised Tokuhashi score lower than 9, associated with an expected survival of less than 6 months. The demographic and clinical characteristics are reported in Table 1.

A total of 13 patients had multiple vertebral metastases; in 11 cases, they were contiguous, and in 2 cases, they were not contiguous. If the lesions were contiguous, they were all included in the stabilization. In cases of distant lesions, we only treated the vertebra with pathological fractures or instability.

Five patients underwent previous surgery and presented with local recurrence. One of them was still alive after 27 months. The other four patients survived 8, 30, 36, and 40 months, respectively, after revision surgery. The relevant clinical data for this patient group are shown in Table 2.

A 52-year-old woman with squamous cell carcinoma metastases in T8 presented local recurrence with spinal cord compression 8 months after debulking, not followed by radiotherapy, and underwent surgical revision. The second surgery was complicated by dural injury, without cerebrospinal fluid fistula. The patient experienced postoperative neurological recovery and was able to walk with walking aids but died 18 months after the surgical revision. In total, 4 patients underwent minimally invasive procedures, and 25 patients underwent posterior decompression and stabilization. We also performed 26 debulking procedures followed by stabilization. En bloc spondylectomy with a double surgical approach was performed on one patient with isolated T10 metastasis from squamous cell carcinoma after multidisciplinary evaluation. The patient died after 27 months from the progression of the neoplastic disease, with no signs of local recurrence.

Nine surgeries (16%) presented one or more complications, which are summarized in Table 3. Postoperative infection was the most frequent complication. We recorded five systemic complications in three patients. A 56-year-old woman operated on for isolated L3 metastasis presented with very serious septicemia with pulmonary and cerebral septic embolism, resulting in tetraparesis, and died 70 days after surgery.

Most patients underwent preoperative embolization in order to block vascular afferents and reduce blood loss during surgery.

We compared the preoperative clinical status (described using Frankel and Karnofsky scores and ambulatory status) with that at the time of hospital discharge, finding a global improvement in all parameters. The details are shown in Table 4a–c.

Neurological function improved in seven (70%) patients with a preoperative Frankel score of C and in seven (64%) patients with a preoperative Frankel score of D. Neurological function remained unchanged in 31 (91%) patients with a preoperative Frankel score of E. The ambulatory status and the Karnofsky score were improved postoperatively. Of the 53 patients with available data, 31 were discharged with crutches or without aids and a general status of over 70. Only 3 patients were bedridden after hospital discharge, compared to 18 patients bedridden before surgery (Figure 2).

We did not consider it appropriate to present a comparison with the clinical status at the last follow-up due to the high number of missing values. The cases in which complete clinical data were available at the last outpatient follow-up still showed further improvement. Overall survival analysis was performed for 56 patients. Forty-eight of them died from the disease (85.7%). The median survival time was 8.1 months (95% CI: 4.7–13.7). The OS rates at 6 months, 1 year, and 3 years were 60% (95% CI: 45.9–71.5), 39.7% (95% CI: 26.7–52.3), and 17.9% (95% CI: 8.8–29.5), respectively (Figure 3).

In total, 20 patients (35.7%) lived for more than 12 months, and 15 patients (26.8%) lived for more than 24 months. All patients who lived for more than 12 months underwent postoperative chemotherapy (traditional or target). Among the patients who lived for more than 12 months, 10 had a Tokuhashi score of less than 9 before surgery. Among these, there were two long survivors, a woman and a man with Tokuhashi scores of 5, alive at 113 and 66 months, respectively, after surgery, both on targeted therapy. Molecular typing was performed on 21 samples of spinal metastasis biopsies using NGS and showed that, in two cases, the EGFR gene was mutated, with two mutations in one case and three mutations in the other case. These patients received targeted therapy with the tyrosine kinase inhibitor osimertinib and had a long survival.

As shown in Table 5, 6 cases of the 21 samples analyzed showed TP53 mutation, and 5 cases showed KRAS mutation. In two cases, both mutations were present. Those patients survived less than 2 years. Only one of them with KRAS mutation responded to targeted therapy and was alive after 5 years.

Thirteen patients died within three months of surgery (22–90 days). Ten of them presented with a status of disseminated disease with bone and/or visceral metastases. Only one patient presented with a Tokuhashi score higher than 9. No other metastases or neurological deficits in progress were identified. As described above, the course was complicated by a very serious septicemic situation resulting in tetraparesis and pulmonary embolism, which led to death after two months.

The univariate analysis of our data (Table 6) provide few answers, though age under 65 and female sex are potentially prognostic factors for overall survival. The other parameters examined do not reach statistical significance. However, the Kaplan–Meier estimates suggest better survival in patients with a postoperative Frankel score of E and a Tokuhashi score ≥9 and in patients who underwent postoperative radiotherapy or chemotherapy.

In Figure 4, we report and illustrate a case of metastasis from adenocarcinoma at the L4 level, treated using debulking and stabilization 30 months after the diagnosis of the primary tumor. After surgery, the patient received targeted chemotherapy with osimertinib and radiation therapy and lived for 30 months with a good quality of life (ambulatory status: autonomous, Frankel score E, Karnofsky score 80 at last follow-up).

## 4. Discussion

The spine is the most frequent site of skeletal metastases from lung cancer [13]. Autopsy findings reveal that 30–70% of lung cancer patients present with spinal metastases [14]. They are typically osteolytic and symptomatic lesions, responsible for the rapid worsening of the disease and associated with a significantly shorter OS [4,15,16].

The treatment of vertebral metastases from lung cancer is a challenge, as the need to achieve local control of the disease by addressing the problems caused by skeletal-related events stands in contrast to the frailty of patients, who may be negatively impacted by an invasive surgery and are often burdened with complications. In fact, for a pathology with a poor prognosis, it is important to carefully evaluate the general clinical status and expected survival. It is important to avoid deterioration caused by a temporary immune deficit following surgery, which can compromise further treatment options. The choice of therapeutic pathway is always multidisciplinary, based on careful individual clinical evaluation. Given the poor prognosis of this disease, many clinicians prefer a conservative treatment based on radiotherapy for local control of the disease [2,17]. However, the indications for surgical treatment are known and commonly accepted: intractable pain, pathological fracture, vertebral instability, and spinal cord and/or radicular compression with neurological impairment. Among lung cancers, SCLC undoubtedly has the fastest course and the worst prognosis. In 2021, Truong et al. [18] retrospectively examined 87 patients surgically treated for lung cancer vertebral metastases, 5 of them with a diagnosis of SCLC. According to their results, the histotype of the primary tumor did not affect postoperative survival. In our series, no surgically treated patient was affected by SCLC. To date, in the few cases of patients with SCLC and symptomatic vertebral metastasis proposed for multidisciplinary evaluation, systemic dissemination and a lower life expectancy have contraindicated surgery.

The choice of treatment and preoperative planning do not only depend on life expectancy but also on the local extension of the disease, neurological status, and the loss of autonomy caused by the metastasis itself. There are numerous prognostic scoring systems in the literature used to guide therapeutic choices. We used a therapeutic algorithm developed by our group in 2004 and validated in 2008, which takes into account the local and systemic conditions and focuses on the importance of a multidisciplinary approach [10,11].

In this study, we also classified the patients using the Tokuhashi scoring system, one of the most widely used in the literature for prognostic studies, in order to understand whether survival prediction based on this score effectively reflects postoperative survival. Truong et al. [18] reported a median survival of 3.8 months in patients with a high Tokuhashi score (<9) and 19.6 months in those with a moderate Tokuhashi score [9,10,11]. In 2017, Igarashi et al. [19] described four cases with neurological impairment for spinal cord compression and Tokuhashi scores ranging from 7 to 9, having a median survival after decompression and stabilization of 42.5 months. In a study published in 2015 by Ha Kee-Yong et al. [20], the Tokuhashi score did not correlate with OS.

In our study, the different survival rates after surgery in the two groups (Tokuhashi score > 9 and Tokuhashi score ≤ 9) did not reach statistical significance, probably due to the small sample size. On the other hand, the median survival was 7.8 months in the first group and 24.3 months in the second, defining a trend that generally confirms the reliability of the assessment scale. All patients who survived for less than three months had a score below 9, except for one patient, whose postoperative course was extremely complicated. However, a detailed analysis of the patients who survived for more than twelve months showed no difference in the preoperative score, which was lower than 9 in half of these cases. Some very long survivors presented with a score that should have contraindicated surgery. In our opinion, this depends on two elements. First, the Tokuhashi score does not take into account recent therapeutic advances. All lung cancers, regardless of the histotype and genomic profile, receive a score of 0 for the parameter corresponding to the primary cancer site. The overall score is often very low, and it can be misleading if strictly used as a prognostic index. Additionally, we have often observed that the resolution of spinal cord compression and the associated neurological deficits drastically improve patient autonomy, allowing for easier access to treatments and indirectly prolonging survival. Therefore, patients with a poor prognosis always have low Tokuhashi scores, but a low Tokuhashi score does not mean that a patient has a poor prognosis. In accordance with the observations of Kobayashi [21] and Igarashi [19], we believe that the Tokuhashi score is not always suitable for predicting survival in lung cancer. In this study, the median survival was 8.1 months, and the mean survival 16 months (range 21 days to 113 months), which can be explained by the relatively high number of patients who survived for more than 24 months (23.2%).

These data are more encouraging than those in other comparable studies. Truong et al. [13] reported a median postoperative survival of 4.1 months in 87 patients. Goodwin et al. [22] described a median survival of 3.5 months in 26 surgically treated patients. Both authors observed that surgery generally led to a clear relief from symptoms and improvement in autonomy, concluding that, despite the poor prognosis, surgical treatment should always be taken into consideration, especially in cases of bed rest due to pain, instability, or spinal cord compression. Conversely, Zairi et al. [17], who reported a median survival of 2.1 months in a study of 53 patients, believe that surgery is not to be recommended for lung cancer metastases. Our data agree with those reported by Ha Kee-Yong [20] (median survival 8.9 months for 25 patients) and Kobayashi [21] (7.5 months on 10 patients).

In 2020, Amelot et al. [16] published a prospective study based on a French multicenter registry. Following the diagnosis of vertebral lung cancer metastases, the median survival of 818 patients was 5.9 months, and it was influenced by general status, neurological condition, and the presence of EGFR gene mutation but not by any surgical treatment. No information was reported on the pre- or post-treatment clinical status. The search for factors that can affect survival has yielded few significant results. Being male and over the age of 65 are negative prognostic factors according to Riihimaki et al. [4]. Other authors, such as Beaufort et al. [23] and Goodwin et al. [22], reported shorter survival in older patients but found no gender differences. In the prospective study of Amelot et al. [16], the survival of patients with lung cancer spinal metastases was not influenced by gender or age.

All the other factors we examined did showed no statistically significant differences, although for some of them, a rather marked trend was observed. Patients without neurological deficits after surgery and those who underwent postoperative chemotherapy and radiotherapy survived longer. It is likely that these observations would be significant with a larger sample of patients. Oncogenic mutations within the EGFR kinase domain are well-established driver mutations in non-small-cell lung cancer (NSCLC). Small-molecule tyrosine kinase inhibitors (TKIs) specifically targeting these mutations have improved treatment outcomes for patients with this subtype of NSCLC. The selectivity of these targeted agents is based on the location of the mutations within the exons of the EGFR gene, and a better structural understanding will inform continued therapeutic development and further improve patient outcomes [24]. We were able to analyze 21 samples of spinal metastasis biopsies via NGS and observed two cases harboring EGFR mutations, which were associated with long-survivor patients treated using targeted therapy with osimertinib.

In our series, 16% of patients presented with one or more surgical complications, an incidence comparable to that reported in other series [17,18]. Complications generally did not affect postoperative survival (Table 4). A relevant aspect that arises from this study is the improvement of all clinical parameters obtained with surgery: neurological status, performance status, and ambulatory autonomy. The latter is the parameter that, above all, correlates with quality of life and the maintenance of activities of daily living (ADLs). This is because bed rest and the inability to walk can be caused by a neurological deficit, a pathological fracture, severe instability, intractable pain, or all the SREs that are indications for surgery. When the surgical indication is correct, a clinical improvement can be achieved regardless of survival time after surgery.

In our experience, local control of the disease can also be obtained in the case of local recurrence. In five patients operated on for relapse in our study, we observed a clear clinical improvement, neurological recovery in cases of deficit, and longer survival compared to our median. Therefore, in the therapeutic decision-making process, it is important not to underestimate the expected survival, given that improving quality of life is the goal of surgery for cancer metastases, and improving autonomy can provide a patient with easier access to cancer treatments, sometimes affecting their life expectancy.

The major limitations of this study are the small sample size and the retrospective data collection. Only patients with symptomatic spinal metastases undergoing surgery were included in this study, in the absence of a control group, and we had incomplete information on the types of cancer treatments performed. A prospective multicenter study with a non-surgical control sample could be useful for overcoming these limitations.

## 5. Conclusions

The results of this retrospective study conducted on a cohort of 56 patients suggest that the surgical treatment of symptomatic spinal metastases from lung cancer improves quality of life, even in patients with a shorter life expectancy, by controlling pain and improving autonomy. The aim of surgery is to locally control metastases and improve symptoms due to SREs. In fact, the algorithm developed and used by our team, which is based only on clinical data, remains useful in the process of selecting patients eligible for surgery.

In this study, the median survival was 8.1 months and the mean survival 16 months because of the relatively high number of patients who survived for more than 24 months. These results were more encouraging than those observed in other comparable studies, even if survival was not the direct goal of the treatment of metastases. However, improving autonomy can give a patient easier access to cancer treatments, sometimes affecting their life expectancy.

In particular, we observed two cases of spinal metastases harboring EGFR mutations, which were associated with long-survivor patients treated using targeted therapy with osimertinib.

## Figures and Tables

**Figure 1 diagnostics-13-02093-f001:**
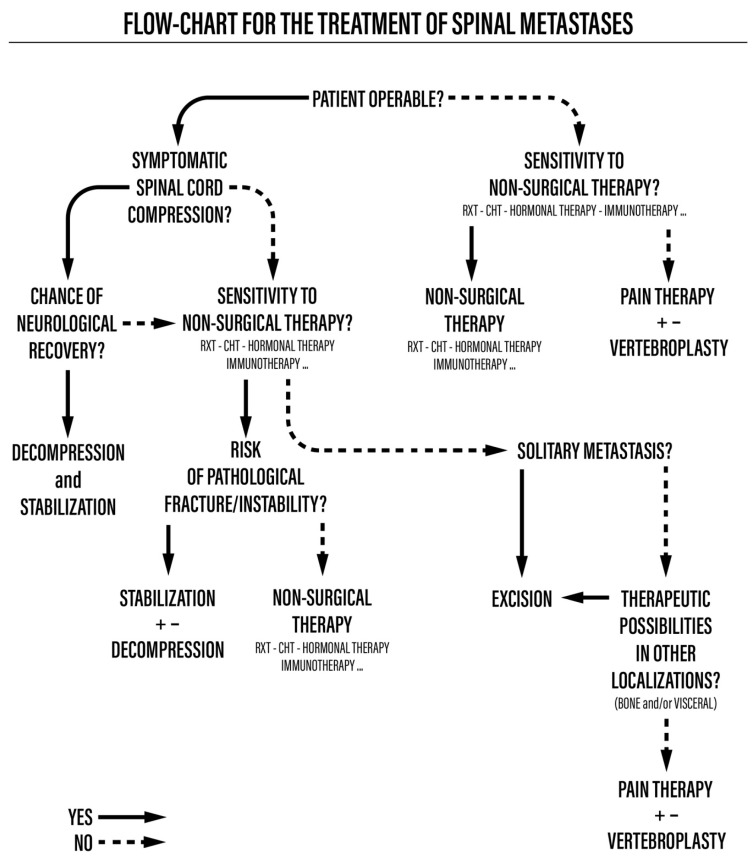
Flowchart for the treatment of spinal metastases.

**Figure 2 diagnostics-13-02093-f002:**
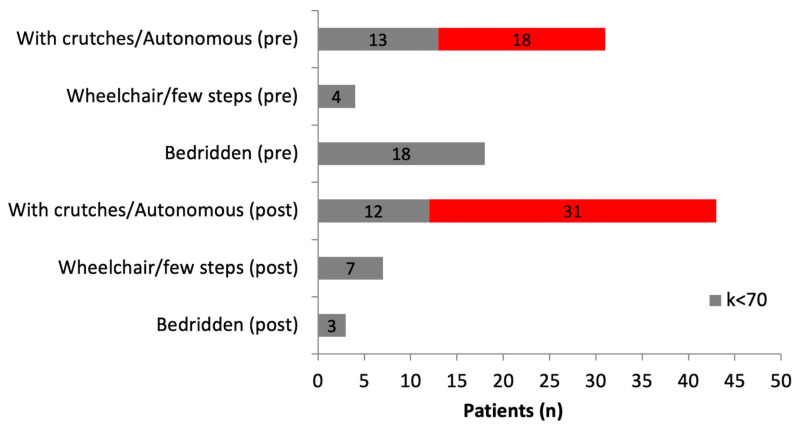
Preoperative and postoperative ambulatory status according to Karnofsky score (3 patients with missing postoperative data were excluded). Grey bar: Karnofsky score < 70; Red bar: Karnofsky score > 70.

**Figure 3 diagnostics-13-02093-f003:**
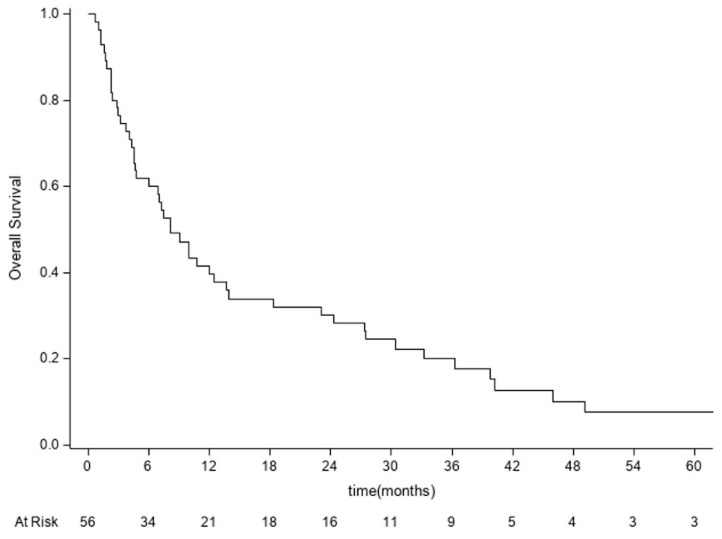
Kaplan–Meier curve for overall survival analysis performed for 56 patients. OS = time from date of surgery to date of death or last follow-up.

**Figure 4 diagnostics-13-02093-f004:**
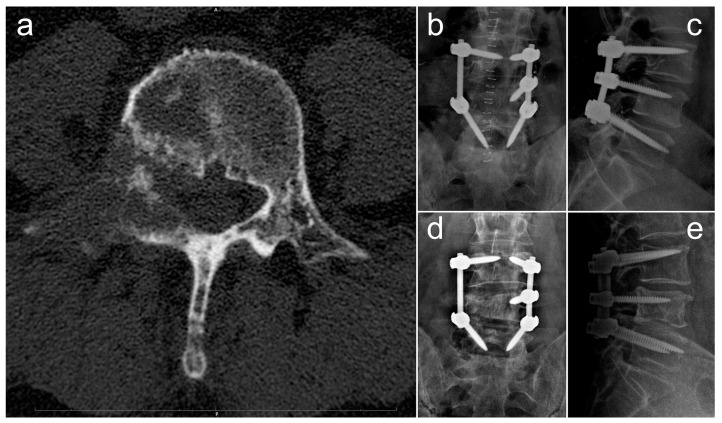
Radiographic images concerning a case of an L4 metastatic lesion from adenocarcinoma, treated via debulking and stabilization. (**a**) Preoperative axial CT scan showing the osteolytic lesion in L4; (**b**,**c**) postoperative anteroposterior (**b**) and laterolateral (**c**) Rx projections showing L3–L5 stabilization; (**d**,**e**) anteroposterior (**d**) and laterolateral (**e**) Rx projections at 18-month follow-up.

**Table 1 diagnostics-13-02093-t001:** Demographic data and clinical characteristics.

	Study Sample (*N* = 56)
**Age**, year median (range)	65 (42–84)
**Sex**, *n* (%)	
Male	34 (60.7)
Female	22 (39.3)
**Levels**, *n* (%)	
Cervical	1 (1.8)
Thoracic	38 (67.9)
Lumbar	17 (30.3)
**Previous treatment**, *n* (%)	
Surgery	1 (1.8)
Surgery + radiotherapy	2 (3.6)
Surgery + chemotherapy	1 (1.8)
Surgery + radiotherapy + chemotherapy	1 (1.8)
Radiotherapy	2 (3.6)
Chemotherapy	9 (16.1)
Radiotherapy + chemotherapy	16 (28.5)
None	24 (42.8)
**Tokuhashi score**, *n* (%)	
0–8	37 (71.1)
≥9	15 (28.9)
**Histology**, *n* (%)	
Adenocarcinoma	37 (66.1)
Squamous carcinoma	9 (16.1)
Neuroendocrine carcinoma	3 (5.3)
Poorly differentiated carcinoma	5 (8.9)
NA	2 (3.6)
**Type of surgery**, *n* (%)	
Debulking	26 (46.4)
Decompression and stabilization	25 (44.6)
Minimal Invasive Spine Surgery (MISS)	4 (7.2)
En bloc resection	1 (1.8)

Note: NA, not available.

**Table 2 diagnostics-13-02093-t002:** Surgeries for local recurrence.

Patient	Man, 59 Years	Man, 52 Years	Man, 71 Years	Man, 63 Years	Man, 76 Years
**Site**	T1	T11	L1	L4	L1
**Histology**	Adeno carcinoma	Adeno carcinoma	Neuroendocrine carcinoma	Adeno carcinoma	Adeno carcinoma
**Surgery**	Decompression–stabilization	Debulking	Debulking	Debulking	MISS
**Complications**	No	No	Pulmonary thromboembolism, cardiac arrest	No	No
**Frankel ^1^ pre > post**	C > E	E > E	E > E	E > E	E > E
**Karnofsky pre > post**	40 > 70	60 > 70	40 > 70	60 > 80	60 > 80
**Therapy post**	Radiotherapy	Unknown	Chemotherapy	Radiotherapy + target therapy	Chemotherapy
**Last follow-up**	40 mo	8 mo	36 mo	30 mo	27 mo
**Last status**	Death	Death	Death	Death	Alive

^1^ Frankel score: A-Complete: No motor or sensory function below the level of lesion; B-Sensory only: No motor function, but some sensation preserved below the level of lesion; C-Motor useless: Some motor function without practical application; D-Motor useful: Useful motor function below the level of lesion; E-Recovery: Normal motor and sensory function, may have reflex abnormalities.

**Table 3 diagnostics-13-02093-t003:** Types of complications.

Complications, *n*/*N* (%)	
Local	7/12 (58.3)
Dural lesion	2/12 (16.7)
Dehiscence/infection	4/12 (33.3)
Postoperative paraplegia	1/12 (8.3)
Systemic	5/12 (41.7)
Pulmonary thromboembolism	2/12 (16.7)
Septicemia	1/12 (8.3)
Intraoperative cardiac arrest	1/12 (8.3)
Transient ischemic attack	1/12 (8.3)

**Table 4 diagnostics-13-02093-t004:** (a) Patients’ neurological status before and after surgery. (b) Patients’ general status before and after surgery. (c) Patients’ ambulatory status before and after surgery.

**(a)**										
**Preoperative** **Frankel Score**		**Postoperative Frankel Score**								
	** *N* **	**B**		**C**		**D**			**E**	
B	1	0		0		1			0	
C	10	0		3		5			2	
D	11	0		1		3			7	
E	34	1		0		2			31	
Total	56	1		4		11			40	
**(b)**										
**Preoperative Karnofsky Score**		**Postoperative Karnofsky Score**								
	** *N* **	**30**	**40**		**50**		**60**	**70**		**80**
30	10	2	3		1		3	1		0
40	9	0	1		5		2	1		0
50	4	0	0		0		1	3		0
60	13	0	0		0		3	10		0
70	14	0	0		1		0	7		6
80	4	1	0		0		0	1		2
Total	54	3	4		7		9	23		8
**(c)**										
**Preoperative** **Ambulatory Status**		**Postoperative Ambulatory Status**								
	** *N* **	**Bedridden**		**Wheelchair/Few Steps**		**With Crutches**			**Autonomous**	
Bedridden	19	2		6		11			0	
Wheelchair/few steps	4	0		0		4			0	
With crutches	9	0		0		5			4	
Autonomous	22	1		1		4			16	
Total	54	3		7		24			20	

(b) Note: 2 patients had missing postoperative Karnofski scores. (c) Note: Two patients’ postoperative ambulatory status was missing.

**Table 5 diagnostics-13-02093-t005:** NGS analysis of gene mutations in 21 cases of lung carcinoma spinal metastases.

Case	EGFR Mutations	Other Mutations
19R000622	WT	KRAS + TP53
19R000570	p.Glu746_Ala750del	/
	p.Thr790Met	
18R002541	WT	KRAS
18R000990	WT	
18R000654	WT	/
17R004012	p.Glu746_Thr751delinsAla	TP53
17R003094	WT	/
17R002628	WT	KRAS
16R003130	WT	/
16R003783	WT	TP53
16_1494	WT	TP53
15_1298	WT	/
13_2838	WT	/
15_1460	WT	KRAS
15_1273	WT	/
13_1140	WT	/
17R00893	WT	/
12_1556	WT	TP53
09_2049	WT	/
19R001520	WT	KRAS + TP53
11_2692C1	WT	/

**Table 6 diagnostics-13-02093-t006:** Univariate analysis of potentially prognostics factors for overall survival.

	Median OS in Months	% 6 m OS	% 1-Year OS	% 3-Year OS	Logrank Test *p* Value
**Age**					
<65 years	11.4 (7.2–27.5)	75% (54.6–87.2)	46.4% (27.6–63.3)	24.1% (10.2–41.1)	0.0561
≥65 years	4.8 (3.2–12.4)	44.4% (25.6–61.7)	32.9% (16.3–50.6)	16.5% (5.3–33)	
**Sex**					
F	18.4 (4.8–39.7)	72.7% (49.1–86.7)	63% (39.4–79.5)	33.9% (15.3–53.7)	0.0170
M	6.9 (3.7–8.1)	51.5% (33.5–66.9)	24.2% (11.4–39.6)	10.9% (3–24.7)	
**Ambulatory status pre**					
Autonomous	18.4 (7.2–30.4)	73.9% (50.9–87.3)	56.2% (33.9–73.6)	19.7% (5.6–40)	0.4641
Bedridden	6.5 (1.7–12)	50% (25.9–70)	27.8% (10.1–48.9)	16.7% (4.1–36.5)	
Other	6.5 (2.3–13.9)	57.1% (28.4–77.9)	28.6% (8.8–52.4)	14.3% (2.3–36.6)	
**Karnofsky pre**					
<70	7.5 (4.5–10)	54% (36.9–68.4)	31.7% (17.5–46.9)	22.7% (10.6–37.5)	0.6132
≥70	13.2 (4.3–27.3)	72.2% (45.6–87.4)	55.6% (30.5–74.8)	14.8% (2.9–35.7)	
**Karnofsky post**					
<70	4.7 (1.8–12)	45.5% (24.4–64.3)	25.6% (9.7–45.1)	20.5% (6.6–39.6)	0.2534
≥70	10.7 (6–27.3)	71% (51.6–83.7)	48.4% (30.2–64.4)	20.7% (8.3–37)	
**Previous surgery**					
Yes	36.3 (8.1–nr)	100%	80% (20.4–96.9)	26.7% (9.7–68.6)	0.1888
No	7.4 (4.5–12)	58% (43.2–70.2)	35.5% (22.6–48.7)	16.5% (7.6–28.2)	
**Frankel pre**					
other	4.7 (2.2–12)	45.5% (24.4–64.3)	27.3% (11.1–46.4)	18.2% (5.7–36.3)	0.4519
E	10.7 (6.9–27.3)	69.7% (51–82.4)	48.1% (30.4–63.8)	20.6% (8.3–36.8)	
**Frankel post**					
altro	6.5 (1.7–12)	50% (24.5–71)	25% (7.8–47.2)	12.5% (2.1–32.8)	0.1083
E	9.1 (4.7–27.3)	64.1% (47–76.9)	45.9% (29.8–60.5)	23.1% (11–37.8)	
**Tokuhashi**					
<9	7.8 (3.7–10.7)	52.8% (35.5–67.4)	29.6% (15.7–45)	20.7% (9.2–35.4)	0.1998
≥9	24.3 (6.9–30.7)	86.7% (56.4–96.5)	66.7% (37.5–84.6)	16.7% (2.9–40.2)	
**Visceral metastases**					
No	9.5 (6.9–13.9)	69% (52.7–80.5)	42.9% (27.8–57.1)	20.4% (9.6–34)	0.7991
Yes	7 (1–nr)	50% (15.2–77.5)	33.3% (5.6–65.8)	16.7% (8.7–50.8)	
**Bone metastases**					
No	12.4 (7–24.3)	75.9% (55.9–87.7)	51.7% (32.5–67.9)	16.1% (5.4–31.9)	0.3795
Yes	4.8 (2.2–9.1)	43.5% (23.3–62.1)	24.8% (9.5–43.8)	24.8% (9.5–43.8)	
**Complications**					
No	9.1 (4.7–13.9)	63% (47.5–75.2)	40.8% (26.6–54.6)	17.6% (8.1–30.2)	0.8234
Yes	6.9 (1.6–36.3)	55.6% (20.4–80.5)	33.3% (7.8–62.3)	16.7% (1.1–49.3)	
**Type of surgery**					
Debulking	10 (4.6–27.3)	68% (46.1–82.5)	48% (27.8–65.6)	24% (9.8–41.7)	0.6993
Decompression and stabilization	6.9 (4–10.7)	52% (31.2–69.2)	31.1% (14.4–49.5)	16.7% (4.8–34.7)	
other	12 (0.7-nr)	60% (12.6–88.2)	40% (5.2–75.3)	20% (0.8–58.2)	
**Radiotherapy pre**					
No	9.1 (3.7–23.1)	58.8% (40.6–73.2)	43.8% (26.8–59.5)	18.8% (7.7–33.6)	0.8279
Yes	8.1 (4.5–13.7)	61.9% (38.1–78.8)	33.3% (14.9–53.1)	21.4% (6.5–41.9)	
**Radiotherapy post**					
No	9.1 (2.8–18.4)	65.2% (42.3–80.8)	43.5% (23.3–62.1)	19.6% (6–38.9)	0.1615
Yes	12.4 (6–39.7)	75% (50–88.7)	54.2% (30.3–73)	32.5% (13.5–53.3)	
**Chemotherapy pre**					
No	10.4 (4.6–23.1)	65.4% (44–80.3)	46.2% (26.6–63.6)	18.5% (6.3–35.5)	0.7500
Yes	7.1 (4–18.4)	53.6% (33.8–69.8)	35.1% (18.7–52.5)	18.2% (6.2–35.3)	
**Chemotherapy post**					
No	8.6 (1.2–18.4)	57.1% (28.4–78)	35.7% (13–59.4)	21.4% (5.2–44.8)	0.1187
Yes	13.9 (7–30.4)	75.9% (55.9–87.7)	54.8% (35.1–70.7)	28.4% (13.1–45.8)	

Nr: not reached.

## Data Availability

Data supporting reported results can be retrieved asking to the corresponding author.

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
