# Peer review of "Indications and Limits of Surgery for Spinal Metastases Derived from Lung Cancer: A Single-Center Experience"

_diagnostics, 2023, doi:10.3390/diagnostics13122093_

Round 1

Reviewer 1 Report

The authors in their article described 56 patients with lung cancer and spinal metastasis 

In a retrospective method, it is well written however required the following 

1- regarding the location of the spinal Mets , is their multiplicity whether contiguous or not and how they deal with 

2- usually Metastic spinal lesions are highly vascular 

The authors should report the vascularity of such pathogy and if they required special blood or plasma transfusions 

3- figure for the only patient presented 

MRI image of patient preoperatively should be uploaded 

I do not see any kind of bone fusion at post operative  X Ray 

The authors should report what type of fusion they use as fixation alone with such pathogy has a high rate of lossening and system failure 

4- heterogeneity of surgical treatment from minimal invasive to maximum debunking and fusion 

Does it affect the rate of comications and the outcome 

Average English editing 

Author Response

Reviewer 1

Comments and Suggestions for Authors

The authors in their article described 56 patients with lung cancer and spinal metastasis. In a retrospective method, it is well written however required the following:

  • regarding the location of the spinal Mets, is their multiplicity whether contiguous or not and how they deal with 

We observed that 13 patients had multiple vertebral metastases; in 11 cases they were contiguous and in 2 cases they were not contiguous. If the lesions were contiguous they were all included in the stabilization, in case of distant lesions we treated only vertebra with pathological fracture or instability.

We added this consideration in the Results section (marked in red).

  • usually Metastatic spinal lesions are highly vascular 

The authors should report the vascularity of such pathology and if they required special blood or plasma transfusions

Considering the vascularity of metastatic spinal lesions, most of our patients underwent pre-operative embolization in order to block vascular afferents and reduce blood loss during surgery.

We added this consideration in the Results section (marked in red).

3- figure for the only patient presented 

MRI image of patient preoperatively should be uploaded 

Pre-operative MRI was not available in our digital archive.

I do not see any kind of bone fusion at post- operative X Ray 

The authors should report what type of fusion they use as fixation alone with such pathology has a high rate of loosening and system failure

In case of surgical treatment of vertebral metastases fusion is not the goal, because the aim of surgery is to locally control the disease, improving the neurological status and the quality of life of patients.  

We performed only primary stabilization with rods and screws without using bone graft, because the life expectancy was low and hardware failures were not detected during the short FU period. Moreover, the use of bone graft can interfere with the visualization of local recurrence which is relevant for oncologic patients.

4- heterogeneity of surgical treatment from minimal invasive to maximum debulking and          fusion 

Does it affect the rate of complications and the outcome? 

We had only 4 cases of minimally invasive surgery (7.2%) and 1 case of en bloc resection (1.8%). Most of patients were treated by debulking (46.4%) and decompression + stabilization (44.6%).

6/12 complications occurred in patients treated with debulking and 6/12 complications occurred in patients treated by decompression + stabilization. The clinical outcomes (Frankel score, Karnofski score and ambulatory status) were also independent of the type of surgery.

Reviewer 2 Report

The authors propose a retrospective cohort of 56 patients with NSCLC related spinal metastases with the aim to assess the overall survival.

The study is very well executed, however with the limitation related to the retrospective design.

Please provide clear conclusions regarding the survival of your study patients within the Conclusion. The readership would expect to see this as it was outcome of interest.

Minor:

Line 50-51: Please revise as it appears that the NSCLC has an affinity to bone, rather then lung. Please emphasize metastatic localization.

There are several minor spelling and grammar mistakes.

Author Response

Reviewer 2

Comments and Suggestions for Authors

The authors propose a retrospective cohort of 56 patients with NSCLC related spinal metastases with the aim to assess the overall survival.

The study is very well executed, however with the limitation related to the retrospective design.

Please provide clear conclusions regarding the survival of your study patients within the Conclusion. The readership would expect to see this as it was outcome of interest.

 As suggested, we added conclusions regarding the survival of our study patients in the Conclusion section (marked in red).

Minor: 

Line 50-51: Please revise as it appears that the NSCLC has an affinity to bone, rather than lung. Please emphasize metastatic localization.

As suggested, we specified that the most frequent localization of NSCLC metastases is skeletal localization.
